# Hypoxia Modulates Transmembrane Prostatic Acid Phosphatase (TM-PAP) in MCF-7 Breast Cancer Cells

**DOI:** 10.3390/ijms26051918

**Published:** 2025-02-23

**Authors:** Marco Antonio Lacerda-Abreu, Luiz Fernando Carvalho-Kelly, José Roberto Meyer-Fernandes

**Affiliations:** Instituto de Bioquímica Médica Leopoldo de Meis, Universidade Federal do Rio de Janeiro, Rio de Janeiro 21941-901, Brazil; marcoantoniolacerdaabreu@gmail.com (M.A.L.-A.); lfernandock@gmail.com (L.F.C.-K.)

**Keywords:** ectophosphatase, hypoxia, prostatic acid phosphatase, MCF-7 cells

## Abstract

In MCF-7 breast cancer cells, transmembrane prostatic acid phosphatase (TM-PAP) plays a critical role in tumor progression, particularly under hypoxic conditions. In this study, the impact of hypoxia on ectophosphatase activity in MCF-7 cells was examined, and the underlying biological mechanisms that influence the breast cancer microenvironment were explored. Compared with normoxic cells, hypoxic cells presented significant reductions in ectophosphatase activity, indicating that hypoxia altered dephosphorylation processes critical for tumor growth and metastasis. Specific decreases in the hydrolysis of substrates, such as p-nitrophenylphosphate (pNPP) and adenosine monophosphate (AMP), were observed under hypoxic conditions, suggesting that hypoxia impaired TM-PAP activity. Further investigation revealed that hypoxia induced an increase in the concentration of reactive oxygen species (ROS), such as hydrogen peroxide (H_2_O_2_), which inhibited ectophosphatase activity. This effect was reversed by the introduction of ROS scavengers. Additionally, hypoxia activated protein kinase C (PKC), further modulating ectophosphatase activity in MCF-7 cells. Collectively, these findings enhanced the understanding of the mechanisms through which hypoxia could influence enzyme activity associated with cancer progression and provide valuable insights into the development of targeted therapeutic strategies.

## 1. Introduction

Breast cancer is one of the most common malignancies and the second most notable cause of cancer-related mortality in women worldwide [1]. Susceptibility to breast cancer is assessed through multiple factors, including environmental, physical, and genetic factors. Breast cancer exhibits significant heterogeneity in phenotype, prognosis, and molecular characteristics, which influence tumor progression, therapeutic response, and patient outcomes [1,2]. This variability underscores the need to investigate specific tumor subtypes and their microenvironmental interactions to improve targeted therapies.

Pathophysiological conditions characterized by hypoxia usually develop at the center of solid tumors because of their rapid development and limited oxygen delivery [3]. Studies on breast cancer have demonstrated that the microenvironment may play a role in the malignant course of the disease [4]. Hypoxia is the most common microenvironmental feature of breast cancer, and it actively encourages metastatic states [5,6]. However, the metastatic machinery involved in the hypoxia response remains mostly unclear, and comprehensive explanations for the development of therapeutics are needed.

Numerous biological and pathological processes involve reactive oxygen species (ROS). ROS levels control the proliferation, invasion, and death of cancer cells and are often elevated in these cells [7]. Hypoxia increases the levels of ROS, such as hydrogen peroxide (H_2_O_2_) and superoxide anion radical (O_2_^−^), from various sources, including electron transport within mitochondria, Nicotinamide Adenine Dinucleotide Phosphate (NADPH) oxidase, and xanthine oxidase [7]. Superoxide is produced when electrons from complexes I, II, and III escape into molecular oxygen via the mitochondrial electron transport chain. Superoxide is then quickly transformed into H_2_O_2_. Because of its stability and capacity to diffuse across membranes, H_2_O_2_ plays a crucial role in signaling [8,9].

The tumor microenvironment contains phosphorylated molecules, such as nucleotides or phosphoproteins [10,11,12]. These molecules are dephosphorylated on the cell surface by ectophosphatases or ectonucleotidases that are released into the extracellular environment, resulting in Pi through various enzymes: (a) ectonucleoside triphosphate diphosphohydrolase (E-NTPDase), which catalyzes the dephosphorylation of nucleotide triphosphate (NTP) and nucleotide diphosphate (NDP); (b) ectonucleotide pyrophosphatase/phosphodiesterases (E-NPPs), which catalyze NTP dephosphorylation, and NPP2, which catalyzes adenosine triphosphate (ATP), adenosine diphosphate (ADP), adenosine monophosphate (AMP), and pyrophosphate (PPi) dephosphorylation; (c) ecto-5-nucleotidase (E-5NT), which catalyzes nucleotide monophosphate (NMP) dephosphorylation; (d) alkaline phosphatase (ALP), which catalyzes NTP, NDP, NMP, and PPi dephosphorylation; and (e) acid ectophosphatase, which catalyzes the dephosphorylation of phosphorylated molecules (P-molecules) and phosphorylated proteins (P-proteins) [13].

Several scholars have investigated the roles of ectonucleotidases and ectophosphatases in modulating the tumor microenvironment, particularly in breast cancer [14,15,16,17]. Recently, Lacerda-Abreu et al. [16] demonstrated that acid ectophosphatase activity contributes to ectonucleotidase function in luminal A MCF-7 breast cancer cells, particularly in AMP hydrolysis and extracellular Pi level regulation. However, the regulatory mechanisms governing ectophosphatase activity in response to key tumor microenvironmental factors remain poorly understood.

In contrast, Lacerda-Abreu et al. [16] focused on characterizing the enzymatic properties of acid ectophosphatase under hypoxic conditions in MCF-7 cells. In previous studies, scholars extensively documented the impact of hypoxia on ectonucleotidases [18,19,20,21,22,23]; however, its influence on ectophosphatase activity remains unexplored.

In this study, we aimed to mimic pathological hypoxia and its physiological effects on the ectophosphatase activity levels of MCF-7 cells. We demonstrated that, compared with normoxic cells, MCF-7 cells cultured under hypoxic conditions presented lower ectophosphatase activity. Furthermore, we sought to elucidate the transduction pathway involved in the modulation of ectophosphatase by hypoxia in MCF-7 breast cancer cells and its role in the biological mechanisms that alter the breast cancer microenvironment.

## 2. Materials and Methods

### 2.1. Materials

Reagents were purchased from Sigma Chemical Co. (St. Louis, MO, USA). All the solutions were prepared with Milli-Q water (Millipore Corp., Bedford, MA, USA).

### 2.2. Cell Culture and Hypoxic Induction

The breast cancer cell line (MCF-7) was grown at 37 °C in a humidified atmosphere of 5% CO_2_ in Dulbecco’s modified Eagle’s medium (Sigma–Aldrich, St. Louis, MO, USA) supplemented with sodium bicarbonate, 10% fetal bovine serum (FBS) (Cripion Biotechnology, Andradina, SP, Brazil), 100 U/mL penicillin, and 100 U/mL streptomycin (Thermo Fisher, SP, Brazil). Before the experiments, the cells harvested from the culture medium were washed two times with buffer consisting of 116 mM NaCl, 5.4 mM KCl, 5.5 mM glucose, 0.8 mM MgCl_2_, and 50 mM HEPES (pH 7.2). Adherent cells were dissociated after incubation at 37 °C in a 5% CO_2_ atmosphere with a trypsin solution (2.5 g/L, pH 7.2, 0.05 mL/cm^2^), and the cell number and protein concentration after Pi treatment were assessed by Neubauer chamber counting and Bradford quantification, respectively [24].

To induce hypoxia, 5 × 10^4^ MCF-7 cells/well were placed into the hypoxia chamber. CO_2_ was absorbed with breath chalk, and the O_2_ partial pressure (pO_2_) was monitored continuously. The isobaric N_2_O/O_2_ mixture was injected with oxygen until it contained 20.9 vol.% oxygen (equivalent to atmospheric oxygen levels) within 7 min. The pO_2_ level was decreased to 5 vol.% and held constant for 10 min, after which the cells were incubated for 60 min with the corresponding treatments.

### 2.3. Ectophosphatase Measurements

Phosphatase activity was determined via the use of *p*-nitrophenylphosphate (*p*NPP) as the substrate and by measuring the rate of *p*-nitrophenol (*p*-NP) production. MCF-7 cells were cultivated in 96-well plates (5 × 10^4^ cells per well) and incubated at 37 °C in a 5% CO_2_ atmosphere in a reaction mixture (0.1 mL) containing 116 mM NaCl, 5.4 mM KCl, 5.5 mM glucose, 50 mM HEPES (pH 7.2), and 0.8 mM MgCl_2_. Reactions were started by adding 5 mM *p*NPP, and the supernatant was transferred to a 96-well plate without cells. After 60 min, the reactions were stopped by adding 0.2 mL of 1 N NaOH, and the results were determined spectrophotometrically at 425 nm [25,26]. Phosphatase activity was calculated by subtracting the nonspecific pNPP hydrolysis measured in the absence of cells [14,15,16]. To determine the concentration of released *p*-NP, a p-NP curve was plotted and used as a standard. For secreted phosphatase, a reaction mixture without pNPP was preincubated for 60 min under normoxic and hypoxic conditions, and 5 mM pNPP was added to the cell-free supernatant [27,28]. We tested nucleotides, such as ATP, ADP, and AMP, as possible ectophosphatase substrates. In these cases, the hydrolytic activities were spectrophotometrically analyzed by measuring the amount of inorganic phosphate (Pi) released from these substrates under the same conditions employed above [29]. Equivalent values were obtained for *p*-nitrophenylphosphatase activity using both methods.

### 2.4. Cell Viability

Cell viability was assessed via a Live/Dead^TM^ Viability/Cytotoxicity Kit (1:20; Invitrogen). MCF-7 cells subjected to hypoxia for 1 h were washed with a reaction mixture (0.5 mL) containing 116 mM NaCl, 5.4 mM KCl, 5.5 mM glucose, 0.8 mM MgCl_2_, and 50 mM HEPES (pH 7.2). For dead cell positive controls, MCF-7 cells were incubated with 100% methanol for 30 min. Images were acquired with an EVOS fl fluorescence microscope from AMG.

### 2.5. H_2_O_2_ Production Assay

The release of H_2_O_2_ produced by the breast cell lines was determined by the Amplex red oxidation (Invitrogen) fluorometric method [30]. MCF-7 cells (1 × 10^6^ cells/mL) were added to a reaction medium containing PBS plus 5 mM glucose, 10 µM Amplex Red, and 0.1 U/mL horseradish peroxidase (HPR) to create a final volume of 0.2 mL. Fluorescence was monitored at excitation and emission wavelengths of 563 ± 5 nm and 587 ± 5 nm for 30 min at 25 °C. H_2_O_2_ concentration was determined using a standard curve with known H_2_O_2_ concentrations.

### 2.6. Thiobarbituric Reactive Substance (TBARS) Measurements

The thiobarbituric acid assay is the most frequently used method for determining the extent of membrane lipid peroxidation in vitro. TBARS production by MCF-7 cells was quantified according to the method proposed by Buege and Aust [31]. The absorbance of the sample was determined at 535 nm against a blank that contained all the reagents without the lipid [15,25].

### 2.7. Protein Kinase C Activity Assay

MCF-7 cells (5 × 10^4^ cells/well) were washed twice in 116 mM NaCl, 5.4 mM KCl, 5.5 mM glucose, 50 mM HEPES (pH 7.2), and 0.8 mM MgCl_2_ and were subsequently lysed in 50 mM Mes-buffer supplemented with 0.1% Triton X-100. PKC activity was assayed in the presence of 4 mM HEPES-Tris, pH 7.0, 0.4 mM MgCl_2_, 1 mM CaCl_2_, 0.36 mg/L neurogranin (a specific substrate for PKC), 25 nM ATP, and 50 g lysed cells to create a final volume of 50 µL in MTS-11C mini tubes (Axygen Scientific, Union City, CA, USA). The reaction was triggered by adding 40 µL of the Kinase-Glo luminescent kit, and, after holding the mixture at 37 °C for 10 min, it was placed in a luminometer (GloMax®-Multi Jr Single-Tube Multimode Reader, Promega Corporation, Madison, WI, USA).

### 2.8. Statistical Analysis

All experiments were performed in triplicate, with similar results obtained from at least three separate cell suspensions. The values presented in all the experiments are the means ± SEs. Differences were considered significant at *p* < 0.05 according to one-way analysis of variance (ANOVA) with Tukey’s multiple comparison test, unless otherwise specified in the figure legends. The kinetic parameters (apparent K_m_ and V_max_ values) were calculated using nonlinear regression analysis of the data for the Michaelis-Menten equation. Linear regression analyses of Lineweaver-Burk plots were performed. All statistical analyses were performed with GraphPad Prism 6.0 software (GraphPad Software, San Diego, CA, USA).

## 3. Results

Recently, we showed that acid phosphatase on the membrane surface of breast cancer cells (MCF-7) is correlated with the tumor process [16]. Since tumors rapidly exhaust the available oxygen in the area, resulting in a hypoxic environment [32], we aimed to investigate the effect of 1 h of hypoxia on ectophosphatase activity in MCF-7 cells. Ectophosphatase activity is significantly decreased by hypoxia (Figure 1A). To investigate the possibility of p-NPP being hydrolyzed by secreted soluble enzymes, MCF-7 cells were incubated in the absence of p-NPP under normoxia and hypoxia for 1 h. The suspensions were subsequently centrifuged to remove the cells, and the supernatants were assayed for phosphatase activity. No differences can be detected in the activity of secreted phosphatases between normoxia and hypoxia (Figure 1A). Under all the conditions tested, 1 h of hypoxia does not decrease MCF-7 cell viability according to the live/dead assay (Figure 1B).

Previously, Lacerda-Abreu et al. [16] demonstrated that prostatic acid ectophosphatase (TM-PAP) activity in MCF-7 breast cancer cells can hydrolyze pNPP and AMP but not ATP or ADP. In this study, the hydrolysis characteristics of pNPP, AMP, ADP, and ATP were investigated in MCF-7 cells under normoxia and hypoxia (Figure 2). Only pNPP and 5′AMP hydrolyses were inhibited by hypoxia, suggesting that hypoxia modulates transmembrane prostatic acid phosphatase activity in MCF-7 cells.

To evaluate the effects of hypoxia on the kinetic characteristics of ectophosphatase activity, we performed pNPP hydrolysis at various pNPP concentrations (0–3 mM) under normoxia or hypoxia (Figure 3A). A Michaelis-Menten kinetic profile of pNPP hydrolysis under normoxia or hypoxia was generated, and reduced V_max_ and the same K_m_ values were obtained under hypoxia (Figure 3A,B, Table 1). Ectophosphatase in MCF-7 cells is more effective under acidic conditions than under other conditions [16]. Therefore, pNPP hydrolysis was assessed at various pH values (5.0–8.0) under normoxia or hypoxia. Compared with that under normoxic conditions, hydrolysis at an acidic pH is considerably lower under hypoxic conditions. These data suggest that hypoxia-regulated ectophosphatase is an acid phosphatase. As shown in Figure 3D, reoxygenation of MCF-7 cells with 20.9% O_2_ for 1 h reverses the effect of hypoxia on ectophosphatase activity, suggesting that hypoxia does not cause irreversible damage to TM-PAP activity.

The release of ROS is one of the primary mediators of hypoxia in breast cancer cells [33,34]. As H_2_O_2_ is a pleiotropic signaling molecule, we aimed to investigate whether H_2_O_2_ could be involved in the cellular events triggered by hypoxia. Thus, H_2_O_2_ production was measured in MCF-7 cells under normoxia or hypoxia. Figure 4A shows that hypoxia induces H_2_O_2_ production in MCF-7 cells. To test whether H_2_O_2_ is necessary for the inhibition of ectophosphatase activity, increasing concentrations of H_2_O_2_ were added to the reaction mixture, and ectophosphatase activity was measured (Figure 4B). Ectophosphatase activity is inhibited at a concentration of 100 μM H_2_O_2_.

As H_2_O_2_ is a versatile signaling molecule, we sought to explore whether H_2_O_2_ might play a role in the cellular events initiated by hypoxic conditions. Ectophosphatase activity was assessed under hypoxia in the presence or absence of the ROS scavenger N-acetyl-cysteine (NAC). These findings indicate that NAC does not affect ectophosphatase activity under normoxic conditions. However, the suppression of ectophosphatase due to hypoxic conditions is undone when NAC is present (Figure 4C).

Hypoxia induces H_2_O_2_ production, which is correlated with lipid peroxidation [35]. We evaluated lipid peroxidation in MCF-7 cells subjected to normoxia or hypoxia for 1 h (Figure 5). We observed that hypoxia induces lipid peroxidation in MCF-7 cells (as determined by the increase in TBARS absorbance at 535 nm). These results suggest that hypoxia-induced lipid peroxidation may be a factor contributing to the inhibition of ectophosphatase activity in MCF-7 cells.

Lipid peroxidation, a hallmark of oxidative stress, produces hydroperoxy fatty acids as initial products that modulate cellular signaling pathways; one form of modulation is the direct activation of PKC by these hydroperoxy acids [36]. In the context of hypoxia in MCF-7 cells, the generation of lipid oxidation products can activate PKC, contributing to the observed modulation of ectophosphatase activity. In breast cancer cells (MDA-MB-231), H_2_O_2_ concentration modulates PKC activity in a dose-dependent manner [37]. Hypoxia can modulate PKC activity through the release of ROS. To investigate whether PKC activity is activated under hypoxia, PKC activity was measured under normoxia or hypoxia for 1 h. As shown in Figure 6A, hypoxia significantly stimulates PKC activity in MCF-7 cells. As H_2_O_2_ activates PKC, we sought to evaluate the effect of PKC on the modulation of ectophosphatase activity in the presence of calphostin C (a PKC inhibitor) under normoxia or hypoxia for 1 h (Figure 6B). Although the inhibitor does not modulate ectophosphatase activity under normoxia, calphostin blocks hypoxia-induced inhibition of ectophosphatase activity in MCF-7 cells.

## 4. Discussion

Ectophosphatases are membrane-attached enzymes whose active sites face the external environment. These enzymes catalyze the removal of phosphate groups from various phosphorylated substrates, such as phosphoramino acids [38,39]. In breast cells, acid ectophosphatase activity has been reported (MCF10-A, MCF-7, and MDA-MB-231) [14]. In MCF-7 cells, acid ectophosphatase has recently been associated with transmembrane prostatic acid phosphatase (TM-PAP) [16]. In addition, the role of these cells is related to aggressive processes such as cell adhesion and migration. We demonstrated that in breast cancer cells (MCF-7), ectophosphatase activity is modulated by hypoxia. Hypoxia is a key microenvironmental difference between tumor and normal tissues that is related to a poor clinical prognosis and resistance to therapies in many solid tumors, including breast cancer [32]. Reoxygenation reverses the inhibition of ectophosphatase activity in hypoxia, indicating that damage only occurs under hypoxic conditions. Low ectophosphatase activity modulated by hypoxia on tumor cell surfaces may be an essential factor modulating the tumor microenvironment.

Zimmermann [40] reported that soluble and secreted prostatic acid phosphatase (PAP), an enzyme that has long served as a diagnostic marker for prostate cancer, has a membrane-bound splice variant that exhibits ecto-5′-nucleotidase activity [34]. Recently, we showed that TM-PAP is responsible for ectophosphatase activity in MCF-7 cells and has a peculiar ability to hydrolyze both AMP and pNPP [16]. Hypoxia-modulated ectophosphatase is TM-PAP for two reasons: (1) the reduction in ectophosphatase activity is more prominent in acidic environments, and (2) in terms of phosphohydrolase activity under hypoxia, only the hydrolyses of *p*NPP and AMP, which are the same substrates as TM-PAP, are affected, as described by Lacerda-Abreu et al. [16].

TM-PAP ectophosphatase activity can be biochemically characterized in MCF-7 cells, with K_m_ and V_max_ values of 0.51 ± 0.03 mM *p*NPP and 618.2 ± 11.43 nmol *p-*NP × h^−1^ × mg protein^−1^, respectively [16]. In this work, hypoxia does not significantly modulate the K_m_ of ectophosphatase activity. However, hypoxia significantly diminishes V_max_. Hypoxia possibly triggers the release of an inhibitor or a signaling pathway that promotes inactivation of the enzyme.

One of the main ways in which cells induce ROS release is through hypoxia, as has been reported in breast cancer cells [41]. The increase in ROS in response to hypoxia occurs via the transfer of electrons from ubisemiquinone to molecular oxygen at the Q_0_ site of mitochondrial complex III [33]. In MCF-7 cells, hypoxia treatment (1 h) promotes the increased production of reactive oxygen species [34]. In this work, hypoxia for 1 h stimulates the production of H_2_O_2_. Furthermore, increasing concentrations of H_2_O_2_ inhibit ectophosphatase activity, suggesting that hypoxia modulates ectophosphatase activity by inducing H_2_O_2_ production. Recently, it has been demonstrated that hydrogen peroxide is capable of inhibiting ectophosphatase activity, as decreases in this activity are related to the oxidation of amino acid residues in the breast cancer cell line MDA-MB-231 [15]. As oxidative stress can regulate intracellular phosphatases [42,43,44,45], ectophosphatase activity may be susceptible to the oxidative stress induced by hypoxia.

One hour of hypoxic exposure was selected to capture acute enzymatic responses, which are critical for understanding early tumor microenvironment adaptations. Previous studies have shown that even short-term hypoxia can induce significant metabolic and enzymatic alterations in cancer cells [46,47,48]. Although prolonged hypoxic exposure may reveal additional regulatory mechanisms, our findings provide initial evidence of hypoxia-induced modulation of ectophosphatase activity. Further studies are necessary to explore the long-term effects of hypoxia and its potential implications for tumor progression.

The tumor microenvironment is known to increase Pi concentrations, which can promote lipid peroxidation, as observed in MDA-MB-231 cells [15]. Lipid peroxidation, a consequence of oxidative stress, generates hydroperoxy acids, which can interfere with various cellular processes, including enzyme activities [49]. In this context, it is plausible that lipid peroxidation contributes to the inhibition of ectophosphatase activity under hypoxic conditions. Although hypoxia induces the production of H_2_O_2_, the exact mechanisms linking lipid peroxidation to ectophosphatase modulation remain unclear. It is reasonable to hypothesize that oxidative damage to membrane lipids, along with the accumulation of lipid peroxidation byproducts, can impair ectophosphatase activity in MCF-7 cells. Further investigations are needed to delineate the precise role of lipid peroxidation and its interactions with hypoxia-induced changes in enzyme regulation within the tumor microenvironment.

While we focused primarily on H₂O₂ because of its relative stability and ability to diffuse across membranes, it is important to consider that other ROS may contribute to the observed modulation of ectophosphatase activity. Superoxide and hydroxyl radicals are highly reactive molecules generated within the tumor microenvironment, particularly under hypoxic conditions [5,8,9]. Superoxide, which is produced primarily by mitochondrial electron transport and NADPH oxidases, can serve as a precursor for H₂O₂ via superoxide dismutase (SOD) activity [9]. Although our results indicate that H₂O₂ plays a central role in ectophosphatase inhibition, future studies should explore whether other ROS species participate in this regulatory mechanism, either by directly modifying enzyme activity or by activating secondary signaling pathways involved in tumor progression.

Hypoxia-induced activation of PKC plays a pivotal role in cancer progression by modulating various processes, including metastasis, chemoresistance, and cytoskeletal remodeling [50,51,52,53]. In alveolar epithelial cells, the ROS-PKC pathway decreases Na,K-ATPase activity [54]. This phenomenon occurs because ROS can induce PKC activity once two pairs of zinc fingers are present within the regulatory domain of the PKC structure. H_2_O_2_ and other oxidants can destroy the zinc finger conformation, and autoinhibition is relieved, resulting in a PKC form that is catalytically active in the absence of Ca^2+^ or phospholipids [55]. In breast cancer cells (MDA-MB-231), Pi-induced ROS production regulates Pi transporters, which is related to PKC activation [37]. In this work, increased PKC activity is observed under hypoxia. Adding a PKC inhibitor (Calphostin) can prevent ectophosphatase inhibition under hypoxia. Analysis of the primary sequence of TM-PAP expressed in MCF-7 cells reveals at least five predicted phosphorylation sites specific for PKC, where direct activation presumably occurs (Appendix A). However, further experiments are needed to elucidate the precise modulatory effect of PKC on ectophosphatase activity.

In this study, we aimed to assess how hypoxia affects ectophosphatase activity by comparing the kinetic characteristics of MCF-7 cells under normoxia and hypoxia. Moreover, one hour of hypoxic incubation was shown to increase H_2_O_2_ generation, which oxidizes and inhibits ectophosphatase activity. We further demonstrated that PKC, which is active in hypoxia, could control ectophosphatase. These findings provide fresh insights into the hypoxic tumor microenvironment, where hypoxia-induced ectophosphatase modulation could affect the reduction in Pi and dephosphorylated molecule release.

## 5. Conclusions

Hypoxia significantly downregulates ectophosphatase activity in MCF-7 breast cancer cells, with transmembrane prostatic acid phosphatase (TM-PAP) being the primary enzyme affected. Mechanistically, hypoxia-induced inhibition of ectophosphatase activity is mediated by increased hydrogen peroxide (H₂O₂) levels, suggesting that oxidative stress plays a central role in enzyme modulation. Furthermore, the activation of protein kinase C (PKC) under hypoxic conditions contributes to ectophosphatase inhibition, as pharmacological PKC inhibition restores enzymatic activity. Given the importance of ectophosphatases in extracellular phosphate metabolism, their modulation under hypoxic conditions may have significant implications for cancer progression, metastatic potential, and therapeutic targeting.

## 6. Future Directions

This study provides novel insights into the hypoxia-induced modulation of ectophosphatase activity in MCF-7 cells; however, some limitations should be acknowledged. Since MCF-7 cells constitute a luminal A breast cancer model, it remains unclear whether similar regulatory mechanisms occur in more aggressive subtypes, such as triple-negative breast cancer (TNBC). Additionally, while our findings highlight hydrogen peroxide as a key modulator, other reactive oxygen species, including superoxide and hydroxyl radicals, may contribute to ectophosphatase inhibition under hypoxia. Furthermore, although we identified putative PKC phosphorylation sites in TM-PAP, direct evidence of PKC-mediated phosphorylation requires further validation through phospho-specific analyses. Future studies should investigate these aspects in different breast cancer subtypes and in vivo models to enhance the understanding of ectophosphatase regulation in the tumor microenvironment.

## Figures and Tables

**Figure 1 ijms-26-01918-f001:**
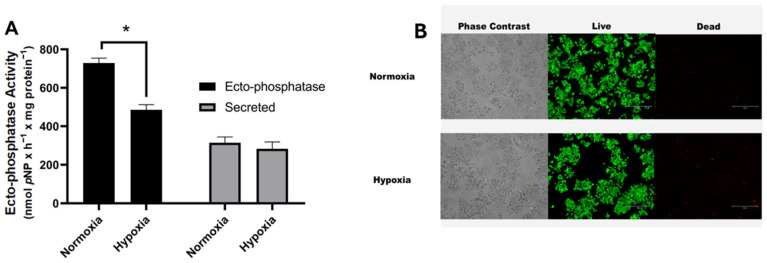
Hypoxia modulates ectophosphatases in MCF-7 cells. (**A**) Intact cells (5 × 10^4^ cells/well = 1.45 mg protein/mL) were incubated at 37 °C, and ectophosphatase activity (dark bars) and secreted phosphatase activity (gray bars) were measured in the reaction medium as described in the Materials and Methods section (Section 2) under normoxia and hypoxia. (**B**) Representative images of MCF-7 cells subjected to hypoxia for 1 h. Corresponding phase contrast images (left panels) and viability images (live/dead assay; green cells are alive, and red cells are dead; right panels). Bars: 300 μm. The data are presented as the means ± SEs from three independent experiments. * *p* < 0.05 indicates significant differences, as assessed by ANOVA followed by Tukey’s multiple comparison test.

**Figure 2 ijms-26-01918-f002:**
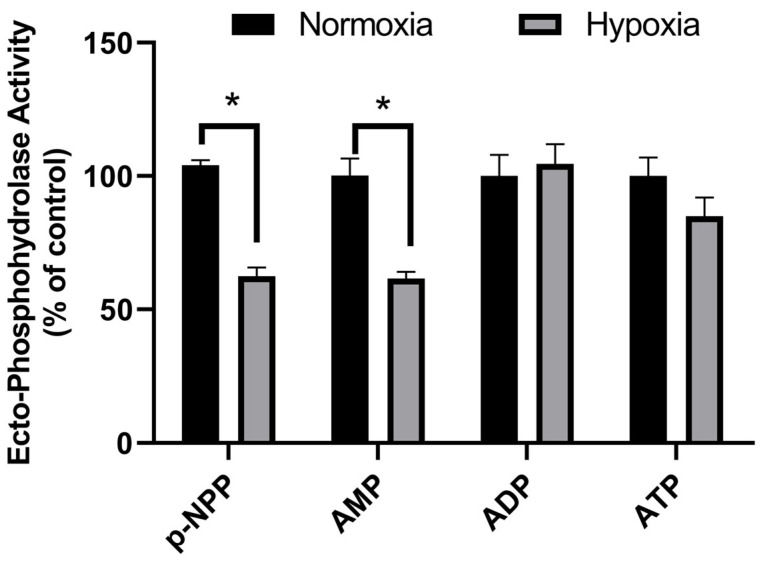
Hypoxia modulation of pNPP and AMP hydrolyses in MCF-7 cells. Intact cells (5 × 10^4^ cells/well = 1.45 mg protein/mL) were incubated at 37 °C, and ectophosphohydrolase activity was measured in the reaction medium as described in the Materials and Methods section (Section 2) under normoxia (black bars) or hypoxia (gray bars) and with 5 mM of one of the following substrates: p-NPP, AMP, ADP, or ATP. The data are presented as the means ± SEs from three independent experiments. * *p* < 0.05 indicates significant differences, as assessed by ANOVA followed by Tukey’s multiple comparison test.

**Figure 3 ijms-26-01918-f003:**
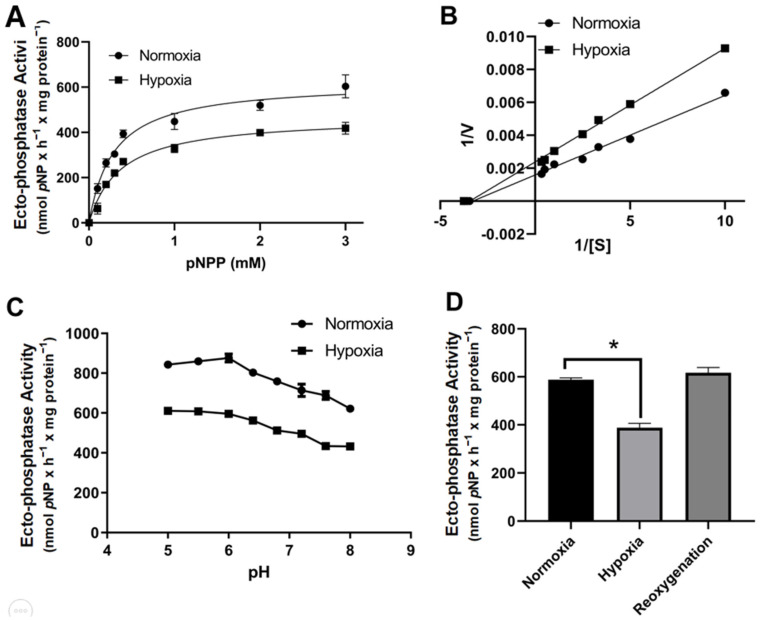
Kinetic parameters of ectophosphatase activity in MCF-7 cells under normoxia and hypoxia. (**A**) Intact cells (5 × 104 cells/well = 1.45 mg protein/mL) were incubated at 37 °C under normoxia or hypoxia, and ectophosphatase activity was measured for 60 min in a reaction mixture (as described in the Materials and Methods section (Section 2)) with increasing concentrations of para-nitrophenylphosphate (p-NPP; 0–3 mM). (**B**) Lineweaver–Burk plot for p-NPP concentrations between 0.1 and 10 mM p-NPP. (**C**) The pH values were adjusted from 5.0 to 8.0 using 15 mM magnesium acetate, 15 mM HEPES, 15 mM Tris, and 15 mM MES. (**D**) Reoxygenation was induced in MCF-7 cells by pretreatment under hypoxia for 60 min, and ectophosphatase activity was subsequently measured for 60 min under normoxia compared with that under hypoxia and control conditions (normoxia). The data are presented as the means ± SEs from three independent experiments. * *p* < 0.05 indicates significant differences from normoxia, as assessed by ANOVA followed by Tukey’s multiple comparison test.

**Figure 4 ijms-26-01918-f004:**
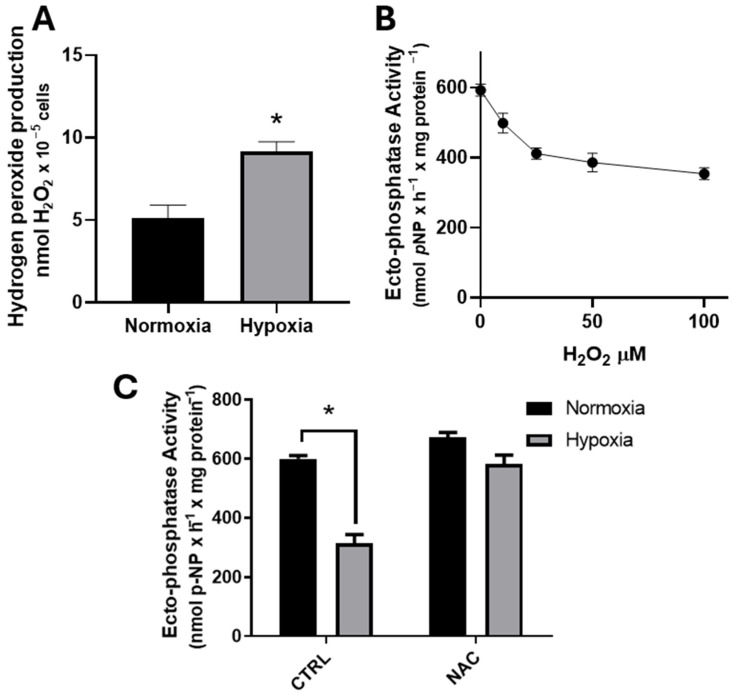
Hypoxia modulates ectophosphatase by inducing H_2_O_2_ production. (**A**) MCF-7 cells (5 × 10^4^ cells/well = 1.45 mg protein/mL) were incubated at 37 °C under normoxia (dark bars) or hypoxia (gray bars) for one hour. H_2_O_2_ production was evaluated at 1 h, as described in the Materials and Methods section (Section 2). (**B**) Ectophosphatase activity was measured for 1 h at different H_2_O_2_ concentrations (0–100 µM) in a reaction mixture as described in the Materials and Methods (Section 2). (**C**) Ectophosphatase activity was measured in MCF-7 cells incubated at 37 °C under normoxic (dark bars) or hypoxic (grey bars) conditions for 1 h, in the absence or presence of N-acetyl-cysteine (NAC).The data are presented as the means ± SEs from three independent experiments. * Denotes significant differences (*p* < 0.05) from normoxia, as assessed by ANOVA followed by Tukey’s multiple comparison test.

**Figure 5 ijms-26-01918-f005:**
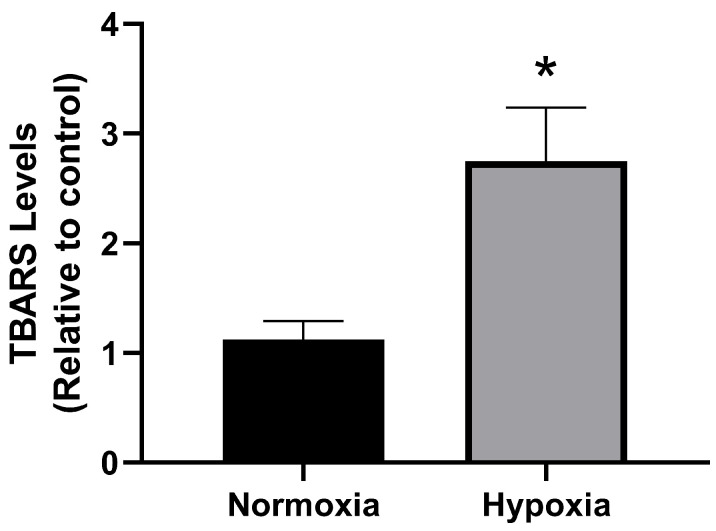
Effect of lipid peroxidation caused by hypoxia. MCF-7 cells (5 × 10^4^ cells/well = 1.45 mg protein/mL) were incubated at 37 °C under normoxia (dark bars) or hypoxia (gray bars) for one hour. Lipid peroxidation induced by TBARS was determined in MCF-7 cells as described in the Materials and Methods section (Section 2). The data are presented as the means ± SEs from three independent experiments. * Denotes significant differences (*p* < 0.05) from normoxia, as determined by an unpaired *t*-test.

**Figure 6 ijms-26-01918-f006:**
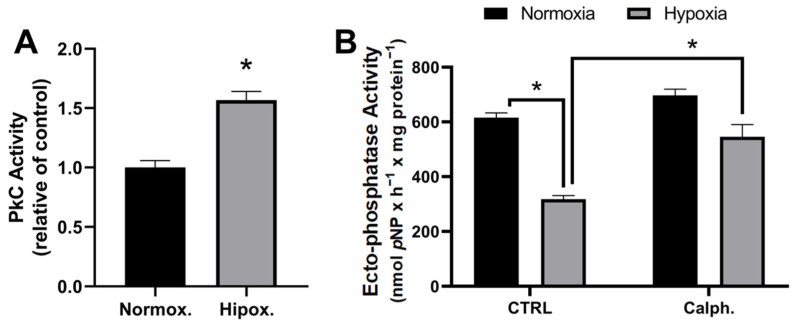
Hypoxia modulates ectophosphatase activity through PKC activity. (**A**) PKC activity was measured as described in the Materials and Methods section (Section 2) in MCF-7 cells preincubated under normoxia and hypoxia for 1 h. (**B**) Ectophosphatase activity was evaluated for 1 h under normoxia and hypoxia in the presence of the PKC inhibitor Calphostin C (Calph: 50 nM) as indicated. The data are presented as the means ± SEs from three independent experiments. * Denotes significant differences (*p* < 0.05) from normoxia, as assessed by ANOVA followed by Tukey’s multiple comparison test.

**Table 1 ijms-26-01918-t001:** Kinetic parameters of p-NPP hydrolysis in normoxia and hypoxia in MCF-7 cells.

Condition	K_m_ (mM pNPP)	V_max_ (nmol p-NP × h^−1^ × mg Protein^−1^)
Normoxia	0.28 ± 0.09	622 ± 60
Hypoxia	0.36 ± 0.10	469 ± 38

Note: Data are presented as the means ± SEs from three independent experiments.

## Data Availability

The data that support the findings of this study are available in the Materials and Methods section.

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
