# Peer review of "Hypoxia Modulates Transmembrane Prostatic Acid Phosphatase (TM-PAP) in MCF-7 Breast Cancer Cells"

_ijms, 2025, doi:10.3390/ijms26051918_

Round 1
Reviewer 1 Report
Comments and Suggestions for Authors
Review comments: I have carefully reviewed the manuscript draft entitled " Hypoxia modulates transmembrane prostatic acid phosphatase 2 (TM-PAP) in breast cancer MCF-7 cells ". In this manuscript draft, the authors examined the impact of hypoxia on ecto-phosphatase activity in MCF-7 cells and explores the underlying biological mechanisms that influence the breast cancer microenvironment. However, it will require a further revision of the manuscript. 1. Lacerda-Abrey et al have reported “Ectonucleotidase Activity Driven by Acid Ecto- phosphatase in Luminal A MCF-7 Breast Cancer Cells” (Cell Biol. Int. 2024). Authors should describe the innovation of this research in the part of “Introduction”. 2. Authors have demonstrated ecto-phosphatase activity was reduced under hypoxic conditions in MCF-7 cells. Authors should consider the ecto-phosphatase activity inother breast cancer cells. 3. The part of “Methods” should be accurately described, Line 92-93, what’s the mean of “20.9 vol.%”? What’s the mean “pOz”, please describe correctly. 4. The figures notes of the manuscript are not standardized, and the experimental methods have been described in the part of "Material and methods", not described in the figures notes. In addition, the description of the figures notes should be consistent with the figures. 5. In Figure 1B, the difference between the two treatment groups is not significant, but there is a significant difference in the detection of phosphorylation. Please explain why? 6. The error value is not correctly expressed in Table 1, and the representation of significant figures is also wrong. 7. Conclusions of the article is missing the part 5, please complete it. 8. The quality of the pictures is poor. Such as Figure 1B. 9. There are few references cited in recent years. 10. There are many places the vocabulary mistakes add up to hamper the reading of the interesting manuscript. I will give only a few short examples: 1) Use the full name when the abbreviation first appears, for example AMP, ADP,… 2) The subscript should be marked correctly, such as “hydrogen peroxide (H2O2)” should change as “H2O2”. In conclusion, in the present form the manuscript should be major revised before publish.
Comments on the Quality of English LanguageThere are many places the vocabulary mistakes add up to hamper the reading of the interesting manuscript.
Author Response
Guest Editor: Prof. Dr. Pawel Sutkowy
International Journal of Molecular Sciences
09/02/2025
Dear Professor Sutkowy,
Please find enclosed the revised manuscript untitled “Hypoxia modulates transmembrane prostatic acid phosphatase (TM-PAP) in MCF-7 breast cancer cells" by Marco Antonio Lacerda-Abreu, Luiz Fernando Carvalho-Kelly and José Roberto Meyer-Fernandes to be considered for publication in the International Journal of Molecular Sciences.
As requested, a new English revision has been completed, and grammatical errors have been corrected throughout the revised manuscript. Consequently, the title has been adjusted to more accurately reflect the scope of the study.
Please find enclosed our reply to the comments contained in your message.
Reviewer 1
I have carefully reviewed the manuscript draft entitled " Hypoxia modulates transmembrane prostatic acid phosphatase (TM-PAP) in breast cancer MCF-7 cells ". In this manuscript draft, the authors examined the impact of hypoxia on ecto-phosphatase activity in MCF-7 cells and explores the underlying biological mechanisms that influence the breast cancer microenvironment. However, it will require a further revision of the manuscript.
Question 1. Lacerda-Abrey et al have reported “Ectonucleotidase Activity Driven by Acid Ecto- phosphatase in Luminal A MCF-7 Breast Cancer Cells” (Cell Biol. Int. 2024). Authors should describe the innovation of this research in the part of “Introduction”.
Answer: As requested, we included the novel contribution of our study in the revised manuscript (Introduction, page 2, lines 63–84). In addition, we included new references in the revised manuscript (Rocha-Vieira et al., 2024; Vignali et al., 2023; Ning et al., 2024; Maralbashi et al., 2024).
Question 2. Authors have demonstrated ecto-phosphatase activity was reduced under hypoxic conditions in MCF-7 cells. Authors should consider the ecto-phosphatase activity in other breast cancer cells.
Answer: We appreciate the reviewer’s suggestion. However, we specifically focused on transmembrane prostatic acid phosphatase (TM-PAP), an acidic ectophosphatase that is exclusively expressed in MCF-7 cells in this study (Lacerda-Abreu et al., 2024). In future studies, we may explore the regulation of other ectophosphatases in different breast cancer subtypes.
Question 3. The part of “Methods” should be accurately described, Line 92-93, what’s the mean of “20.9 vol.%”? What’s the mean “pOz”, please describe correctly.
Answer: We appreciate the reviewer’s comment and have clarified these terms in the revised manuscript (Methods, page 3, lines 111-114).
Question 4. The figures notes of the manuscript are not standardized, and the experimental methods have been described in the part of "Material and methods", not described in the figures notes. In addition, the description of the figures notes should be consistent with the figures.
Answer: As requested by the reviewer, the figure legends were standardized, the methods were kept only in the "Materials and Methods" section, and the figure descriptions were adjusted in the revised manuscript (figure legends, page 5, lines 199–1207, page 5, lines 216-219, page 6, lines 220-223; page 6, lines 240–243, page 7, lines 244-252; page 8, lines 274-283; page 9, lines 293–298; Page 10, lines 319-326; Methods, Page 3, lines 129–131).
Question 5. In Figure 1B, the difference between the two treatment groups is not significant, but there is a significant difference in the detection of phosphorylation. Please explain why?
Answer: The Figure 1B shows the results of a LIVE/DEAD cell viability assay, indicating that hypoxia does not modulate cell viability. In contrast, Figure 1A demonstrates a significant reduction in phosphatase activity under hypoxic conditions. Notably, this decrease in enzymatic activity cannot be attributable to a loss of cell viability, as confirmed by the viability assay.
Question 6. The error value is not correctly expressed in Table 1, and the representation of significant figures is also wrong.
Answer: The error values in new Table 1 and the representation of significant figures were corrected in the revised manuscript (Results, page 6, lines 254–256).
Question 7. Conclusions of the article is missing the part 5, please complete it.
Answer: As requested, the conclusion (part 5) is now included in the revised manuscript (page 12, lines 334–444).
Question 8. The quality of the pictures is poor. Such as Figure 1B.
Answer: As suggested by the reviewer, the quality of each Figure, including Figure 1B, has been improved.
Question 9. There are few references cited in recent years.
Answer: As requested, additional references from the last three years were incorporated in the revised manuscript (New References: Carvalho-Kelly et al., 2023; Carvalho-Kelly et al., 2024; Rocha-Vieira et al. 2024, Vignali et al. 2023, Ning et al. 2024, Maralbashi et al. 2024; Bayar et al., 2024; Bayar and Bildik, 2021; Taze et al., 2022).
Question 10. There are many places the vocabulary mistakes add up to hamper the reading of the interesting manuscript. I will give only a few short examples: 1) Use the full name when the abbreviation first appears, for example AMP, ADP,… 2) The subscript should be marked correctly, such as “hydrogen peroxide (H2O2)” should change as “H2O2”. In conclusion, in the present form the manuscript should be major revised before publish.
Answer: The suggested revisions have been implemented throughout the revised manuscript, including the proper use of full names before abbreviations and the correct formatting of subscripts (page 1, line 17; page 2, lines 52, 66-67; page 8, line 266-267).
Reviewer 2 Report
Comments and Suggestions for Authors
This manuscript has thoroughly studied the effects of hypoxia on ecto-phosphatase activity in MCF-7 breast cancer cells. This was also validated using various molecular mechanisms involved, focusing on the role of reactive oxygen species (ROS), lipid peroxidation, and protein kinase C (PKC) activation in modulating enzyme activity. Overall, it looks like clear hypothesis driven by well-planned objectives. The study does well to link hypoxia-induced ROS generation to changes in ecto-phosphatase activity and further ties these changes to lipid peroxidation and PKC activation, adding depth to the understanding of hypoxia’s impact on the tumor microenvironment. However, manuscript requires substantial changes and need to be improved further in few sections mentioned in the comments below.
Comments to author:
1. Few sentences can be improved for clarity. e.g, “The phenotype, prognosis and molecular hallmarks of breast cancer are heterogeneous [1-2].” can author be more specific, how heterogeneity affects the relevance of the study.
2. Some minor grammatical errors were noted (e.g., "The data are the means ± SEs of three experiments with different cell suspensions" appears frequently and could be streamlined).
3. While the study uses 1-hour hypoxic exposure, it would be valuable to discuss whether this time frame is representative of chronic hypoxic conditions in breast tumors, or if longer exposure might yield more physiologically relevant findings.
4. The study majorly focuses on hydrogen peroxide , but other ROS (such as superoxide or hydroxyl radicals) could also contribute to the observed effects. A brief mention or exploration of the involvement of other ROS could add depth to the findings.
5. While the manuscript hints at PKC activation as a potential mediator in the hypoxia-induced modulation of ecto-phosphatase activity, a more detailed mechanistic explanation is needed. How exactly PKC activation impacts ecto-phosphatase activity could be explored in greater detail, including potential pathways or proteins involved in this process.
6. It would be useful to include a section that acknowledges the limitations of the study and suggests potential future directions. For example, one limitation might be the use of MCF-7 cells, which are hormone receptor-positive, while other breast cancer subtypes (such as triple-negative breast cancer) could yield different results in hypoxic conditions.
Minor comment:
The figure legends are generally good, but a little more detail on experimental conditions (e.g., concentration of reagents, specific conditions) could be added for clarity.
Author Response
Guest Editor: Prof. Dr. Pawel Sutkowy
International Journal of Molecular Sciences
09/02/2025
Dear Professor Sutkowy,
Please find enclosed the revised manuscript untitled “Hypoxia modulates transmembrane prostatic acid phosphatase (TM-PAP) in MCF-7 breast cancer cells" by Marco Antonio Lacerda-Abreu, Luiz Fernando Carvalho-Kelly and José Roberto Meyer-Fernandes to be considered for publication in the International Journal of Molecular Sciences.
As requested, a new English revision has been completed, and grammatical errors have been corrected throughout the revised manuscript. Consequently, the title has been adjusted to more accurately reflect the scope of the study.
Please find enclosed our reply to the comments contained in your message.
Reviewer 2
This manuscript has thoroughly studied the effects of hypoxia on ecto-phosphatase activity in MCF-7 breast cancer cells. This was also validated using various molecular mechanisms involved, focusing on the role of reactive oxygen species (ROS), lipid peroxidation, and protein kinase C (PKC) activation in modulating enzyme activity. Overall, it looks like clear hypothesis driven by well-planned objectives. The study does well to link hypoxia-induced ROS generation to changes in ecto-phosphatase activity and further ties these changes to lipid peroxidation and PKC activation, adding depth to the understanding of hypoxia’s impact on the tumor microenvironment. However, manuscript requires substantial changes and need to be improved further in few sections mentioned in the comments below.
Question 1. Few sentences can be improved for clarity. e.g, “The phenotype, prognosis and molecular hallmarks of breast cancer are heterogeneous [1-2].” can author be more specific, how heterogeneity affects the relevance of the study.
Answer: The sentence has been revised to clarify how breast cancer heterogeneity impacts disease progression and prognosis in the revised manuscript (Introduction, page 1, lines 34–38).
Question 2. Some minor grammatical errors were noted (e.g., "The data are the means ± SEs of three experiments with different cell suspensions" appears frequently and could be streamlined).
Answer: Minor grammatical errors have been corrected throughout the revised manuscript, including streamlining repetitive phrasing for clarity and conciseness.
Question 3. While the study uses 1-hour hypoxic exposure, it would be valuable to discuss whether this time frame is representative of chronic hypoxic conditions in breast tumors, or if longer exposure might yield more physiologically relevant findings.
Answer: We appreciate the reviewer’s consideration regarding the relevance of the 1-hour hypoxic exposure. In response, we have clarified the rationale for selecting this time frame in the revised manuscript (Discussion, page 11, lines 377-384). Additionally, we included new studies in the reference list (Bayar et al., 2024; Bayar and Bildik, 2021; Taze et al., 2022).
Question 4. The study majorly focuses on hydrogen peroxide, but other ROS (such as superoxide or hydroxyl radicals) could also contribute to the observed effects. A brief mention or exploration of the involvement of other ROS could add depth to the findings.
Answer: We appreciate this suggestion. In the revised manuscript, we have included a discussion acknowledging the potential contributions of other reactive oxygen species (ROS) in modulating ectophosphatase activity (Discussion, page 10, lines 398-408).
Question 5. While the manuscript hints at PKC activation as a potential mediator in the hypoxia-induced modulation of ecto-phosphatase activity, a more detailed mechanistic explanation is needed. How exactly PKC activation impacts ecto-phosphatase activity could be explored in greater detail, including potential pathways or proteins involved in this process.
Answer: Thank you for your insightful comment. To address this issue, we have included an additional analysis in the revised manuscript (page 12, lines 420–422). Specifically, we examined the primary sequence of TM-PAP expressed in MCF-7 cells and identified at least five predicted phosphorylation sites specific for PKC, suggesting a potential direct regulatory interaction (New Table S1). These findings support the hypothesis that PKC activation may directly modulate ectophosphatase activity through phosphorylation events.
Question 6. It would be useful to include a section that acknowledges the limitations of the study and suggests potential future directions. For example, one limitation might be the use of MCF-7 cells, which are hormone receptor-positive, while other breast cancer subtypes (such as triple-negative breast cancer) could yield different results in hypoxic conditions.
Answer: As requested by the reviewer, we have included a new section entitled "Future Directions" in the revised manuscript. In this section, we discuss the potential differences in ectophosphatase regulation across breast cancer subtypes, the possible involvement of other reactive oxygen species (ROS), and the need for further validation of PKC-mediated phosphorylation of TM-PAP (page 12, lines 445–457).
Minor comment: The figure legends are generally good, but a little more detail on experimental conditions (e.g., concentration of reagents, specific conditions) could be added for clarity.
Answer: As suggested by the reviewer, we included additional details on the experimental conditions in all the Figure legends.
We hope that in its present form, the manuscript is now acceptable for publication in the International Journal of Molecular Sciences. We are grateful for the thoughtful suggestions made by the reviewers.
Thank you for your consideration.
Sincerely,
Prof. José Roberto Meyer-Fernandes, Ph.D.